# Exploring Changes in Land Use and Landscape Ecological Risk in Key Regions of the Belt and Road Initiative Countries

**Xuebin Zhang [1,2,3,\*], Litang Yao [1], Jun Luo [4]⊙ and Wenjuan Liang [5]**

1. College of Geography and Environmental Science, Northwest Normal University, Lanzhou 730070, China; 2021272832@nwnu.edu.cn
2. Key Laboratory of Resource Environment and Sustainable Development of Oasis, Lanzhou 730070, China
3. Gansu Engineering Research Center of Land Utilization and Comprehension Consolidation, Lanzhou 730070, China
4. College of Resources and Environment, Gansu Agricultural University, Lanzhou 730070, China; luoj@gsau.edu.cn
5. Human Resources Department, Jinchuan Group Thermoelectric Co., Ltd., Jinchang 737100, China; wzl1900566@163.com
* Correspondence: zhangxb@nwnu.edu.cn

**Abstract:** The Belt and Road Initiative (BRI) has revealed that it is necessary to strengthen research on land use and land cover change (LUCC) and ecological risk in key regions of countries around the world. In this study, the spatiotemporal characteristics of LUCC in the five capitals of Central Asian countries within the BRI were analyzed. Based on the grid scale, a landscape pattern index was introduced to quantitatively evaluate the landscape ecological risk levels of the five capitals. The results showed the following: first, the components of land use types in the five capitals have different structural characteristics, which are mainly grassland, unused land, and cultivated land. The landscape types that changed significantly were water and unused land, while the construction land area showed a trend of continuous increase. Second, different capitals have different land-use transfer patterns. Akmola State is mainly converted from cultivated land to grassland; Chuy State is mainly converted from forest land to grassland; Dushanbe and Tashkent City are mainly converted from grassland to forestland; and Ahal State is mainly converted from grassland to unused land. Third, the overall landscape ecological risks of the five capitals were low. Akmola State had the largest proportion of lowest ecological risk areas, whereas Chuy State and Dushanbe City had an increasing trend of highest ecological risk areas. The level of ecological risk in Tashkent remained stable during the study period, and the highest ecological risk areas in Ahal State decreased to 49,227.86 km$^2$. This study has enriched the research results of land use change and landscape ecological risk assessment of countries within the BRI and can provide a research reference for these countries and regions to achieve ecological sustainable development and strengthen ecosystem management.

**Keywords:** land use and cover change (LUCC); ecological risk; spatiotemporal evolution; Belt and Road Initiative (BRI)

## 1. Introduction

The Belt and Road Initiative (BRI) was introduced by China in 2013 with the goal of enhancing regional connectivity through infrastructure projects, such as railways, roads, ports, pipelines, and energy facilities [1]. However, the demand for land and water resources, and the resulting changes in land use and land cover, may have significant negative impacts on landscapes and ecosystems. Thus, this may pose a threat to landscape ecological security and the surrounding environment in countries of the BRI. LUCC plays an important role in influencing the spatial structure of landscape types, intensity of disturbance resistance, landscape diversity, and ecological function [2,3]. The fragmentation of landscape types, the decrease in landscape diversity, the degradation of landscape functions, and the decline

in the ability of landscape to resist external disturbance are closely related to frequent human activities and the intensification of land use change owing to rapid economic development [4,5]. Therefore, the study of spatial patterns and temporal changes in land use and land cover in countries within the BRI is helpful for better understanding the impact mechanism of land use and land cover on landscape ecological security.

The International Earth and Biosphere Program (IGBP) and the International Human Program on Global Environmental Change (IHDP) identified three important LUCC issues: mechanisms of land use change, mechanisms of land cover change, and regional and global models [6]. Until 1999, LUCC research content was supplemented, emphasizing that the mechanism of land use change should pay more attention to its decision-making, change process, pattern evolution, and multi-scenario analysis [7,8]. Current LUCC research results further deepen on the basis of these hotspots, emphasizing the driving factors and contribution degree when revealing the mechanism of land use and land cover change [9]. The LUCC driving forces exist mainly in natural and social systems. The natural systems driving factors are relatively stable over the long term, whereas the human driving factors of social systems are relatively dynamic. Therefore, research has mainly focused on exploring the driving mechanism of the social economy [10–13]. Studies on LUCC evolution mechanisms largely include long-term historical evolution characteristic analysis [14,15] and future evolution simulation [16–18]. Based on quantitative analysis of land-use change rate, dynamic degree, and other indicators [19,20], the focus has been on remote sensing image interpretation and GIS spatial analysis. Obtaining the dynamic evolution and spatial difference of spatial patterns [21,22] and using FRAGSTATS software to calculate the landscape pattern index to describe landscape evolution is also an important part [22–24]. The ecological and environmental effects of LUCC emphasize the impacts of land use change on climate, resource use, ecosystem services, and their value [25–27]. Artificial land use and land cover change often pose a serious threat to ecological security [28,29]; for example, it can lead to intensified landscape fragmentation, reduced connectivity of landscape patterns, and increased ecological risk levels of landscapes, resulting in adverse consequences, such as ecosystem function degradation and biodiversity loss [30,31]. Many studies have quantitatively evaluated the ecological risks caused by human land use change. The landscape loss model was constructed by analyzing the responses of land use/land cover, landscape pattern evolution and landscape ecological processes to natural factors or human disturbance. The product of risk occurrence probability and potential damage is taken as the result of ecological risk assessment [32,33]. With the support of landscape ecology theory, the landscape loss model based on land use data can describe landscape structure quantitatively and explain the evolution mechanism of landscape ecological risk from the perspective of spatial pattern change. Therefore, this model has been widely used [34]. The five Central Asian countries are adjacent to China's Xinjiang province, which is not only an important fulcrum of the BRI, but also the first stop of China's outbound investment. With the proposal and implementation of the BRI, China's investment in Central Asian countries has been increasing and China has become an important investment partner of the five Central Asian countries, focusing on energy, mineral resources and infrastructure construction [35,36]. Since 2013, the total amount of China's foreign direct investment stock in the five Central Asian countries has been an increasing trend on the whole, with a decline in 2015 and 2016. From the perspective of investment flow, Kazakhstan received the largest investment flow, accounting for about 50–60% of the total investment of the five countries, followed by Tajikistan and Kyrgyzstan. In terms of investment stock, China's direct investment stock in the five Central Asian countries exceeded US $14 billion. By the end of 2018, China's investment stock in Kazakhstan was the highest at $7.341 billion, accounting for 50% of China's investment stock in the five Central Asian countries in 2018. Uzbekistan followed with $3.69 billion, accounting for 25.13%. The investment stock in Uzbekistan and Tajikistan showed a steady increase; China has the lowest direct investment stock in Turkmenistan, and by the end of 2018, it was only $312 million, accounting for 2.13% [37–39].

Since the implementation of the BRI, basic research on land use change, landscape pattern evolution and ecological risk assessment in countries along the BRI is still lacking. The five Central Asian countries, as important investment gathering places along the BRI and the largest inland region in Central Asia, are ideal regions for relevant research. As an important national economic and cultural center, the capital is the carrier of various production activities [40]. The capitals of the five Central Asian countries are the economic and industrial centers of the country [41–45]. With the implementation of the BRI, large-scale infrastructure construction projects have been implemented, the urbanization process is accelerating, and the land use pattern and structure are becoming increasingly complex. Meanwhile, these activities also have an important impact on the complexity of the ecological environment. Taking the cities or states where the capitals of five Central Asian countries are located as typical regions, a landscape ecological risk index evaluation model was constructed and applied to characterize and analyze the dynamic changes of land use and landscape ecological risks. This study not only enriched the practical cases of relevant research along the BRI, but also provided scientific basis and theoretical reference for countries along the Belt and Road to formulate effective land-use development and optimization policies, strengthen ecological risk control measures and protect the ecological environment.

## 2. Materials and Methods

### 2.1. Study Area

Generally, the five republics of Kazakhstan, Turkmenistan, Uzbekistan, Tajikistan, and Kyrgyzstan are collectively referred to as the five Central Asian countries [46,47]. The capitals are Nur Sultan (Kazakhstan), Ashgabat (Turkmenistan), Tashkent (Uzbekistan), Dushanbe (Tajikistan), and Bishkek (Kyrgyzstan). The five Central Asian countries are located at the center of the Eurasian continent along the Silk Road Economic Belt, which is the transportation hub connecting the Eurasian continent and the only route of the ancient Silk Road. In 2018, the total land area of these countries was approximately 4 million $km^2$, the total population was about 72.49 million, the urbanization rate was 48.16%, and the GDP was 277.420 billion dollars. Among these countries, Kazakhstan has the largest land area, accounting for 68.09% of the total area. Meanwhile, the per capita GDP of Kazakhstan is the highest, approximately 11.29 times that of Tajikistan. The most populous country in Central Asia is Uzbekistan, accounting for 45.46% of the total population [47].

To better analyze the spatial-temporal changes of land use in the five Central Asian countries within the BRI, capitals with good economic development were selected in this study. Concurrently, the difference in the area and size of the capitals were considered.

In Kazakhstan, Akmola State was selected as the study area, with an area of $15.7 \times 10^4$ $km^2$, with a population of about 0.74 million. Nur Sultan is located in the central part of Kazakhstan, beside the ISIM river. It is the political, economic, trade and tourism center of Kazakhstan [41]. In Kyrgyzstan, Chuy State was selected as the study area, with an area of $2.2 \times 10^4$ $km^2$, with a population of about 0.74 million. Kyrgyzstan mainly attracts foreign investment in the processing industry, scientific fields, mineral exploitation, etc., and the investment is mainly concentrated in the capital Bishkek and Chuy State [42]. In Tajikistan, Dushanbe City was selected as the study area, with an area of $3.2 \times 10^4$ $km^2$. As the capital of Tajikistan and the most densely populated city, Dushanbe is a national transportation hub, a key city for infrastructure construction, and a major inflow of foreign investment [43]. In Uzbekistan, Tashkent City was selected as the study area, with an area of $1.7 \times 10^4$ $km^2$. It is the main concentration of foreign-funded enterprises [44]. In Turkmenistan, Ahal State was selected as the study area, with an area of $10.2 \times 10^4$ $km^2$. As an oasis city, Ahal State has gathered a large number of industrial enterprises in Turkmenistan. In June 2019, the world's first natural gas-to-oil project in cooperation between Turkmenistan and Japan was put into operation in Ahal State. At the same time, Ahal State also has textile, building materials, steel and other industries, and its output value is in the forefront of the country [45]. (Figure 1). The study areas have a temperate continental climate, with

hot summers and cold winters, large annual and daily temperature differences, scarce precipitation, and an arid climate.

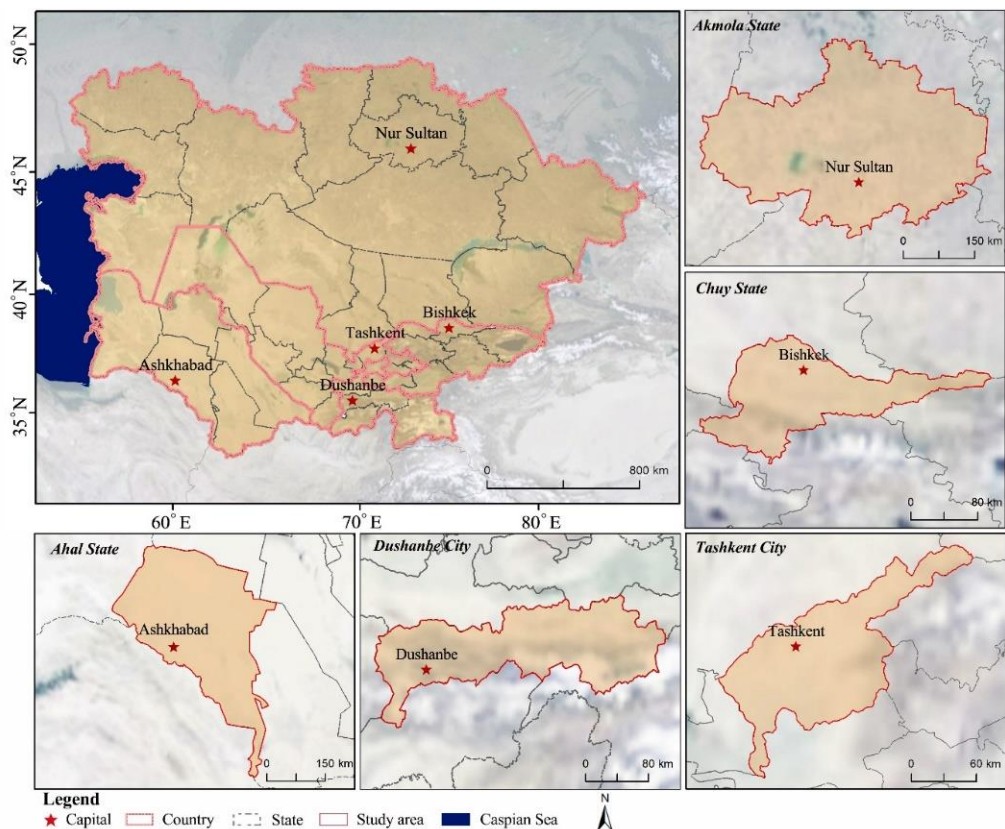

**Figure 1.** Locations of the study area.

## 2.2. Data Sources

This study used three global surface coverage datasets from 2000, 2010, and 2020 (GlobeLand30) (http://www.globallandcover.com/, accessed on 3 December 2021) [48], which is the WGS-84 coordinate system. Data accuracy evaluation was conducted by the Academy of Aerospace Information Innovation, Chinese Academy of Sciences. In 2020, the overall accuracy of the data was 85.72% and the kappa coefficient was 0.82 [49]. GlobeLand30 data include 10 first-level land types: cultivated land, forestland, grassland, shrubland, wetland, water body, tundra, artificial surface, bare land, glacier, and permanent snow cover. Coalesced with the GlobeLand30 classification system and according to the actual situation, the missing land types were eliminated and reintegrated with other land types. Finally, six land use types including cultivated land, forestland, grassland, water, construction land, and unused land were used for analysis. The land use reclassification is shown in Appendix A (Table A1).

## 2.3. Dynamic Degree of Land Use

### 2.3.1. Dynamic Degree of Single Land Use

The dynamic degree of single land use (*D*) reflects the quantity change of a certain land type in a certain time range and can indicate the change speed and change range of different land use types in a certain period [50]. The calculation formula is as follows:

$$D = \frac{U_j - U_i}{U_i} \times \frac{1}{T} \times 100\% \tag{1}$$

where $U_i$ and $U_j$ represent the area (km²) of a certain land use type at the beginning and end of the study period, respectively. $T$ is the study period (typically in y). $D$ is the dynamic degree toward certain land uses during the study period.

2.3.2. Dynamic Degree of Comprehensive Land Use

The comprehensive land use dynamic degree (*LC*) is mainly used to analyze the overall quantity change in land use types in the study area [51]. The calculation formula is as follows:

$$LC = \frac{\sum_{i=1}^{n} \Delta LU_{i-j}}{2 \sum_{i=1}^{n} LU_i} \times \frac{1}{t} \times 100\% \tag{2}$$

where $LU_i$ represents the area of class $i$ land use type at the initial stage of the study, $\Delta LU_{i-j}$ represents the absolute value of $i$ land type converted to other land use types throughout the study period, $t$ represents the study duration.

*2.4. Land-Use Transfer Matrix*

The land-use transfer matrix can reveal the direction and area of land-use type change in a certain period, reflect the conversion relationship between different land use types, and further understand the structural characteristics of different land types before and after transfer [52]. The change in land use type in a certain period can be explained effectively by the transfer matrix. The calculation formula is as follows:

$$S_{ij} = \begin{bmatrix} S_{11} & S_{12} & \cdots & S_{1n} \\ S_{21} & S_{22} & \cdots & S_{2n} \\ \vdots & \vdots & \vdots & \vdots \\ S_{n1} & S_{n2} & \cdots & S_{nn} \end{bmatrix} \tag{3}$$

where $S$ is the area, $n$ is the number of land types, $i$ and $j$ are the land use types at the beginning and end of the study period, respectively. The row elements in the matrix represent the area of $i$ class transferred to another class, and the column elements represent the area of another class transferred to $j$ class.

*2.5. Landscape Ecological Risk Index*

The intensity and internal resistance of the regional ecosystem to external disturbances determine ecological risk. Different landscape types have different resistance to external disturbances [53]. In this study, the landscape ecological risk index was constructed by selecting the landscape disturbance, vulnerability, and loss indices based on relevant literature, and the regional ecological risk level was represented by risk plots.

2.5.1. Landscape Disturbance Index

The landscape disturbance index can reflect the degree of external disturbance to different landscape systems and is directly proportional to the landscape ecological risk [54,55]. In this study, the landscape fragmentation, separation, and dominance indices were superimposed to reflect the degree of landscape disturbance. The calculation formula is as follows:

$$U_i = aC_i + bF_i + cD_i \tag{4}$$

$$C_i = \frac{N_i}{A_i} \tag{5}$$

$$F_i = \frac{\sqrt{S_i}}{2P_i} \left( S_i = \frac{N_i}{A}, P_i = \frac{A_i}{A} \right) \tag{6}$$

$$D_i = dL_i + eP_i \left( L_i = \frac{N_i}{N} \right) \tag{7}$$

where $U_i$ is the index of the landscape disturbance degree, $C_i$ is the fragmentation degree of the landscape type, $F_i$ is the abruption degree of the landscape type, $D_i$ is the dominance degree of the landscape type. According to relevant studies [53–55], $a$, $b$ and $c$ are the weights of the crushing, separation, and dominance degrees, which are 0.5, 0.3, and 0.2, respectively. $L_i$ is the relative density of the landscape type, $P_i$ is the relative coverage of the landscape type, $d$ and $e$ are their weights, which are 0.6 and 0.4, respectively.

### 2.5.2. Landscape Vulnerability Index

The landscape vulnerability index represents the vulnerability of the internal structure of different landscape systems and reflects the resistance of different landscape types to external disturbances. Hence, the greater the vulnerability of the landscape, the lesser the external disturbance of landscape types, and vice versa. Based on relevant studies [52–54], this study classified the vulnerability of landscape types in the study area from high to low: unused land, water, cultivated land, grassland, forestland, and construction land. After the normalized treatment, the vulnerability index $E_i$ of each landscape type was obtained.

### 2.5.3. Landscape Loss Index

The superposition of the landscape disturbance index and the landscape vulnerability index can reflect the degree of loss of different landscape systems [55–57]. The formula for calculating the landscape loss index $R_i$ is as follows:

$$R_i = U_i \times E_i \tag{8}$$

### 2.5.4. Landscape Ecological Risk Index

The calculation formula of landscape ecological risk index is as follows:

$$ERI_i = \sum_{i=1}^{N} \frac{A_{ki}}{A_k} R_i \tag{9}$$

where $ERI_i$ is the ecological risk index of the $i$ risk community, $A_{ki}$ is the area of the $i$ type landscape of the $k$ risk community, $A_k$ is the area of the $k$ risk community, and $R_i$ is the landscape loss index of the $i$ type landscape.

## 3. Results

### 3.1. Characteristics of Land Use in Key Areas within the BRI

By comparing the area proportions of different land use types in five cities or states in 2000, 2010, and 2020, the dominant land use types in each city or state are significant, and grassland, unused land, and cultivated land are the three main types in general. The land use structure of these areas was relatively stable from 2000–2020, and the difference in the proportion of land type area between 2000–2010 and 2010–2020 was not significant (Figure 2). Akmola State is mainly cultivated and grassland, with cultivated areas accounting for 49–50% of the total area. Chuy State is mainly composed of grassland and cultivated land, with grassland areas accounting for 43–45% of the total area. Dushanbe City is mainly composed of grassland and unused land, accounting for 48% of the total area and 30% of the total area, respectively. Tashkent City is mainly cultivated land and grassland, which account for approximately 31–33% of the total area, and the grassland area accounts for approximately 25–27% of the total area. Unused land and grassland are the main areas in Ahal State, accounting for 51–54% of the total area.

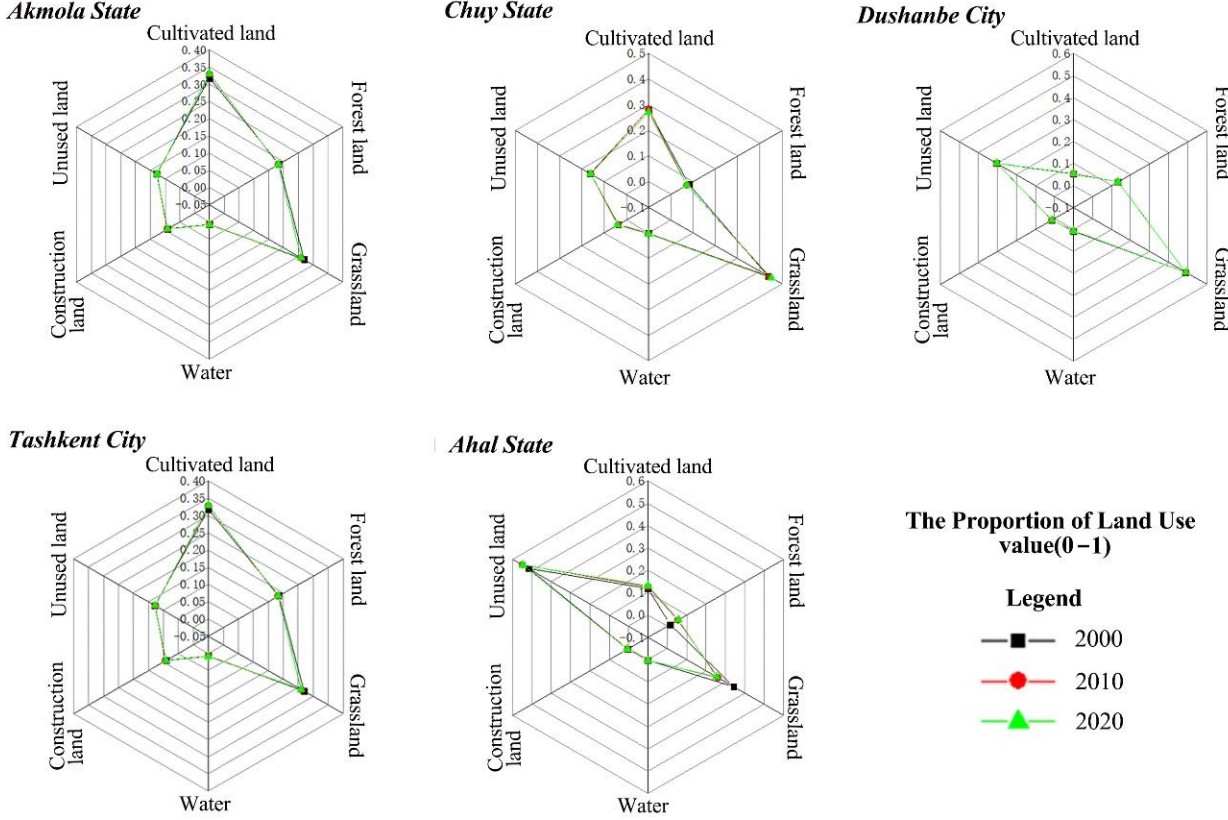

**Figure 2.** Proportion of land use types.

*3.2. Spatial-Temporal Changes of Land Use in Key Areas within the BRI*

During the study period, the range of land use change was largest in Ahal State and smallest in Dushanbe City (Table 1).

**Table 1.** Dynamic degree of integrated land use.

| Study Area | 2000–2010 | 2010–2020 | 2000–2020 |
|---|---|---|---|
| Akmola State | 0.36% | 0.43% | 0.14% |
| Chuy State | 0.17% | 0.21% | 0.21% |
| Dushanbe City | 0.06% | 0.02% | 0.03% |
| Tashkent City | 0.25% | 0.04% | 0.13% |
| Ahal State | 1.47% | 0.17% | 0.88% |

Comparing the dynamic degree changes of comprehensive land use from 2000 to 2010 and from 2010 to 2020, the dynamic degree of comprehensive land use in Akmola State and Chuy State showed an increasing trend. Among them, the dynamic degree toward comprehensive land use in Akmola State increased by 7% from 2010 to 2020. This indicates that the land-use change rate was faster in Akmola State and Chuy State during this period. The dynamic degree of comprehensive land use in Dushanbe City, Tashkent City and Ahal State showed a downward trend, among which Ahal State showed the most obvious decline, indicating that the rate of land development and utilization in Ahal State reduced.

From 2000 to 2020, the spatiotemporal evolution characteristics of different land use types in each study area were different. Simultaneously, in the two research periods of 2000–2010 and 2010–2020, different land use types in the study area had great differences in area change and dynamic degree of single land use (Figures 3–5).

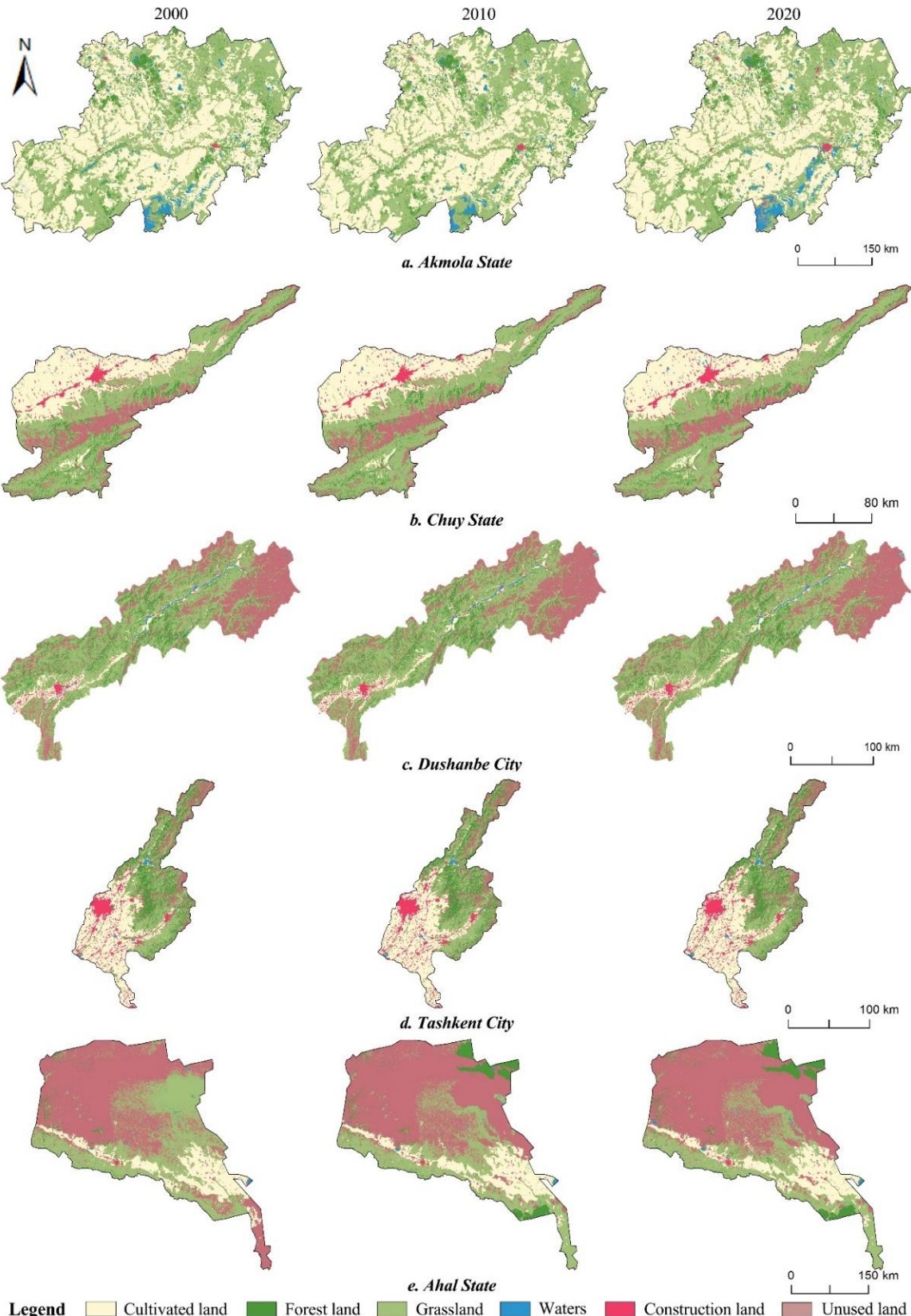

**Figure 3.** (**a**–**e**) Spatial distribution map of land use change.

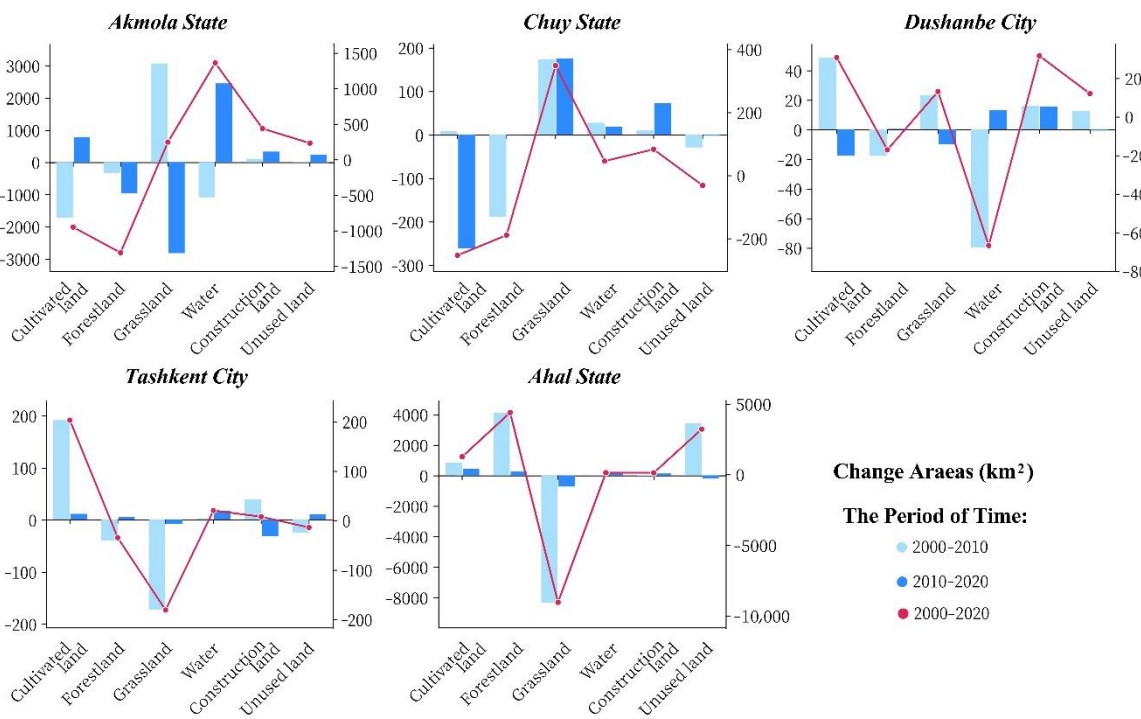

**Figure 4.** Change of land use area in different periods.

|  |  | Cultivated land | Forestland | Grassland | Water | Construction land | Unused land |
|---|---|---|---|---|---|---|---|
| **Akmola State** | 2000–2020 | −0.12 | −1.26 | 0.04 | 3.21 | 3.98 | 10.88 |
| | 2010–2020 | 0.2 | −1.92 | −0.87 | 15.61 | 5.49 | 23.77 |
| | 2000–2010 | −0.43 | −0.66 | 1 | −5.16 | 1.94 | −0.91 |
| **Chuy State** | 2000–2020 | −0.4 | −0.4 | 0.35 | 8.29 | 1 | −0.09 |
| | 2010–2020 | −0.81 | −0.01 | 0.34 | 4.38 | 1.75 | −0.01 |
| | 2000–2010 | 0.03 | −2.05 | 0.35 | 10.01 | 0.23 | −0.16 |
| **Dushanbe City** | 2000–2020 | 0.18 | −0.04 | 0.01 | −2.53 | 0.71 | 0.01 |
| | 2010–2020 | −0.2 | 0 | −0.01 | 1.41 | 0.68 | 0 |
| | 2000–2010 | 0.57 | −0.08 | 0.03 | −6.04 | 0.72 | 0.03 |
| **Tashkent City** | 2000–2020 | 0.37 | −0.11 | −0.39 | 1.58 | 0.05 | −0.06 |
| | 2010–2020 | 0.04 | 0.03 | −0.04 | 2.67 | 0.4 | 0.1 |
| | 2000–2010 | 0.7 | −0.25 | −0.75 | 0.43 | 0.5 | −0.22 |
| **Ahal State** | 2000–2020 | 1.03 | 29.39 | −2.58 | 3.72 | 2.33 | 0.61 |
| | 2010–2020 | 0.67 | 0.96 | −0.55 | 7.67 | 5.4 | −0.08 |
| | 2000–2010 | 1.35 | 55.15 | −4.74 | −0.17 | 0.59 | 1.29 |

**Dynamic attitude of land use(%)** −6.04 ▬▬▬ 55.15

**Figure 5.** Dynamic degree of land use from 2000 to 2020.

The cultivated land area of Akmola State decreased by 953.12 km² from 2000 to 2020 but increased from 2010 to 2020. The total area of forestland decreased by 1312.45 km², with a dynamic degree of −1.26%. Meanwhile, the dynamic degree toward forest land use in 2010–2020 was greater than that in 2000–2010. During 2000–2010 and 2010–2020, the water and unused land areas first decreased then increased. However, the water area decreased greatly during 2000–2010, and the unused land area increased the most during 2010–2020.

Construction land showed a trend of continuous increase, with the largest growth rate from 2010 to 2020 and a dynamic degree of 5.49%.

The cultivated land area of Chuy State increased slightly from 2000 to 2010 but decreased significantly from 2010 to 2020. The area of forest land and unused land shows a continuous downward trend, the reduction range of the two types of land decreases from 2010 to 2020, and the water area shows a continuous increasing trend with a large change range.

The areas of cultivated land, grassland, construction land, and unused land in Dushanbe City increased from 2000 to 2020, but the overall change range was small, and the water area decreased significantly from 2000 to 2010, with a maximum dynamic degree of −6.04%. The water area increased from 2010 to 2020 with a dynamic degree of 1.41%.

The water area of Tashkent City has increased by 67.2% from 2000 to 2020. Both forestland and unused land showed a trend of decreasing first and then increasing. At the same time, the range of change from 2000 to 2010 was greater than that from 2010 to 2020. The grassland area showed a continuously decreasing trend, and the decreasing range became smaller after 2010. The area of construction land showed an increasing trend during the study period.

The grassland area of Ahal State decreased significantly from 2000 to 2010, accompanied by a significant increase in forestland area. This trend persisted between 2010 and 2020, but the dynamic degree decreased significantly. From 2010 to 2020, the water and construction land areas increased significantly. The unused land first increased then decreased from to 2000–2010 to 2010–2020, but the area increased slightly throughout the study period.

### 3.3. Analysis of Land Use Transfer Patterns in Key Areas within the BRI

Through analysis of the land-use transfer matrix, the main conversion relationships between land use types in each study area from 2000 to 2020 were obtained. There were different transfer patterns among land use types in each study area (Figure 6).

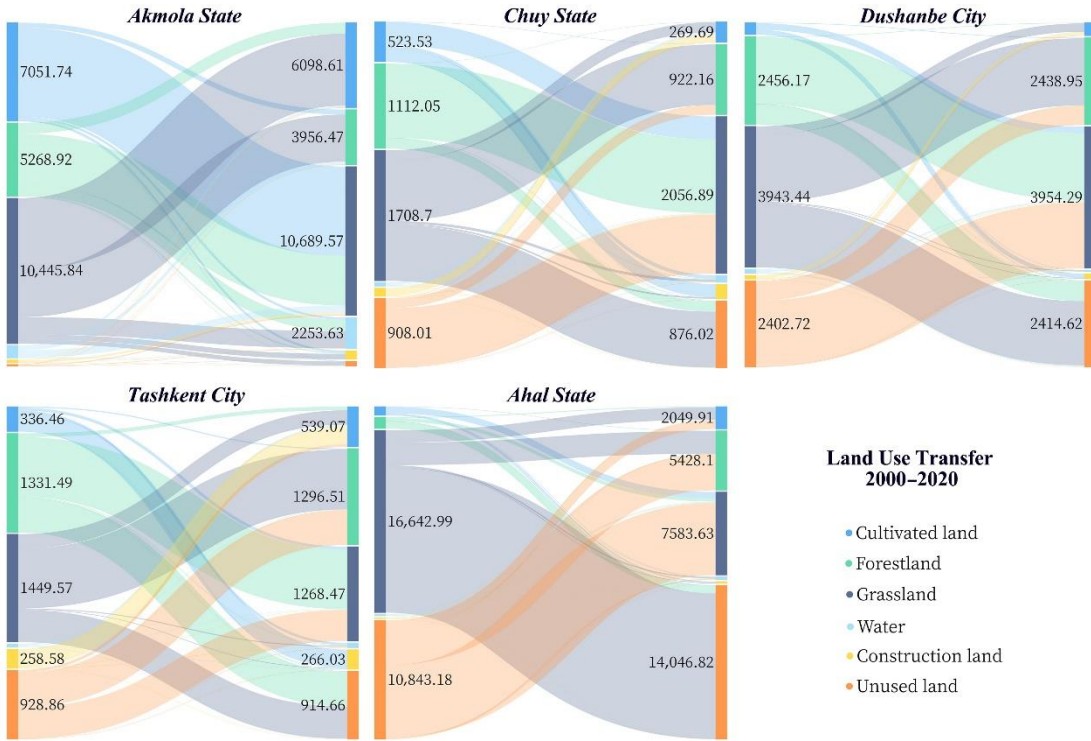

**Figure 6.** The main transfer relationship between land use types.

From 2000 to 2020, the area of grassland transferred out in Akmola State was the largest, and the area converted into cultivated land accounted for 49.49% of the total area of grassland transferred out, while the area from cultivated land to grassland accounted for 90.63% of the total area transferred out of cultivated land; grassland had the largest net transfer area, mainly from cultivated land and forest land. The land-use transfer pattern of Akmola State is mainly from cultivated land to grassland, supplemented by grassland to cultivated land. Chuy State has the largest area of grassland transferred out, mainly into forest land and unused land. The area from forest land to grassland accounts for 86.72% of the total area transferred out of forest land. However, the main land types transferred from grassland are forest land and unused land. Therefore, the land-use transfer pattern in Chuy State is mainly from forest land to grassland, supplemented by grassland to forest land. The land-use transfer pattern of Dushanbe City is the same as that of Chuy State. Tashkent City had the largest area of grassland transferred out, mainly into forest land and land for use, and the net transfer area of forest land was the largest, mainly from grassland and unused land. Thus, the land-use transfer pattern of Tashkent City is mainly from grassland to forestland, supplemented by forestland to grassland. Ahal State has the largest area of grassland net transfer out, and the area converted to unused land accounts for 79.87% of the total area transferred out. Moreover, the net transfer area of unused land is the largest, mainly from grassland and forest land. Hence, the land-use transfer pattern in Ahal State is mainly from grassland to unused land, supplemented by the conversion of unused land to grassland.

### 3.4. Spatial-Temporal Evolution of Landscape Ecological Risk in Key Areas within the BRI

In this study, the landscape ecological risk index was calculated by dividing ecological risk communities, the spatial distribution characteristics of landscape ecological risk were obtained (Figure 7), and the area occupied by each grade of ecological risk zone. Natural breakpoint method was used to classify landscape ecological risk into five levels: lowest risk (ERI ≤ 0.05), lower risk (0.05 < ERI ≤ 0.10), medium risk (0.10 < ERI ≤ 0.15), higher risk (0.15 < ERI ≤ 0.20), highest risk (0.20 < ERI).

Overall, the landscape types of the highest risk and higher risk areas are mainly unused land and water, the vulnerability of unused land and water is high, and the resistance to external environmental disturbance is poor. The medium-risk areas are mainly the interlaced distribution areas of unused land edge and cultivated land along the river, with a high degree of landscape fragmentation. The main landscape types in the lower and lowest-risk areas were cultivated land, forestland, grassland, and construction land.

During the entire study period, the proportion of the lowest ecological risk areas in Akmola State was dominant, showing a slight growth trend, and the areas of other risk areas were reduced to varying degrees (Figures 8 and 9). From 2000 to 2020, Chuy State was dominated by lowest-risk and lower-risk areas. During the study period, the lowest-and medium-risk areas decreased, the lower-risk and higher-risk areas increased, and the highest risk area decreased in 2010 but increased in 2020. Dushanbe City had the largest proportion of medium risk areas in the year 2000. In 2010, the areas of lower risk areas and medium risk areas decreased significantly, while the areas of higher risk areas and highest risk areas increased significantly, with the area of highest risk areas accounting for 58.82%. Throughout the study period, Tashkent City had the largest proportion of lowest-risk areas, followed by the highest risk areas, while the area of lower risk areas decreased during the study period, and the area of other risk areas increased. The increase from 2010 to 2020 is greater than that from 2000 to 2021. In 2000, the area of highest risk areas in Ahal State accounted for the largest proportion, followed by lower risk areas; from 2000 to 2010, the area of medium risk areas increased significantly, and the area of highest risk areas decreased significantly; from 2010 to 2020, the area changes of risk areas at all levels remained basically stable.

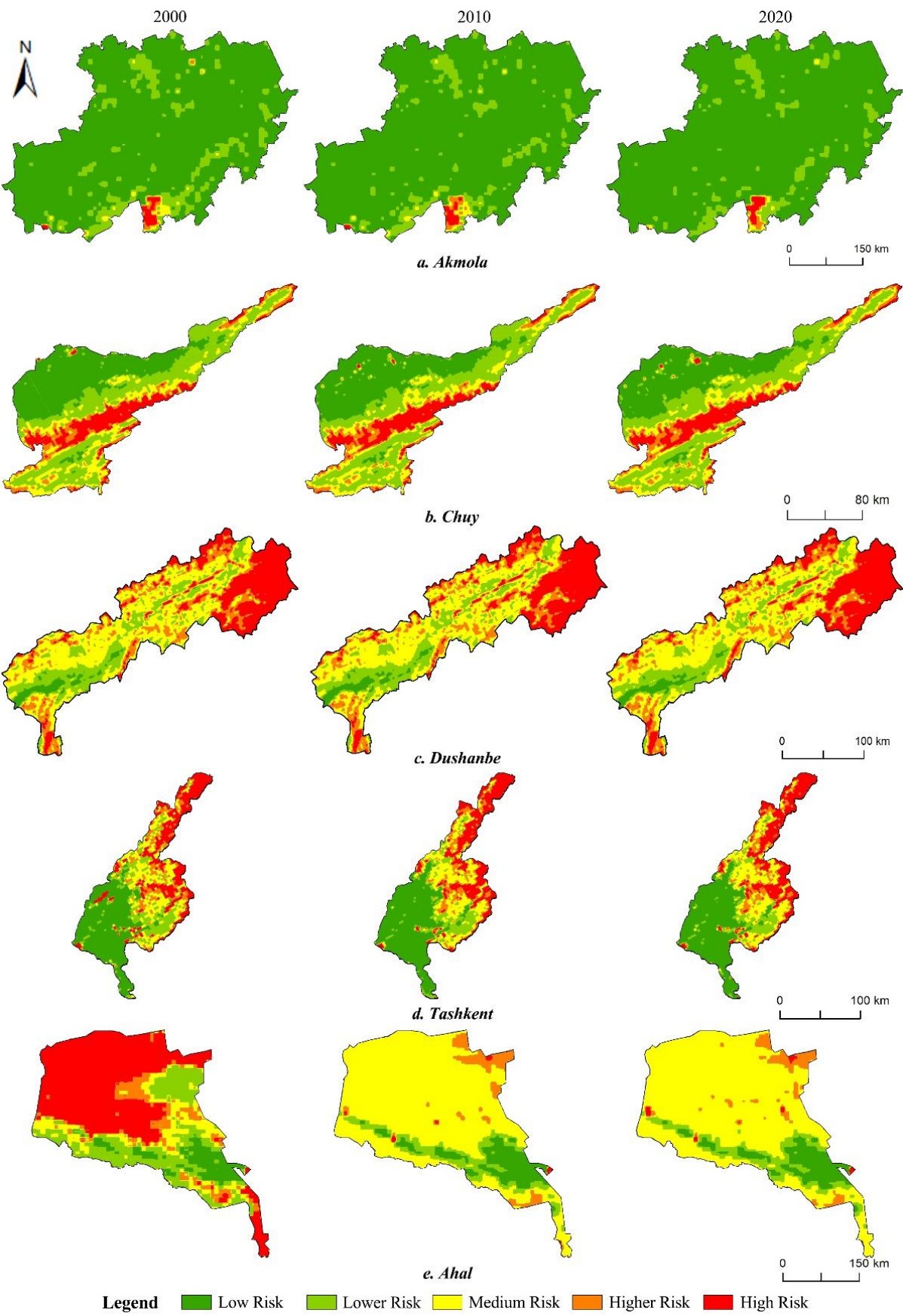

**Figure 7.** (**a–e**) Spatial distribution map of landscape ecological risk.

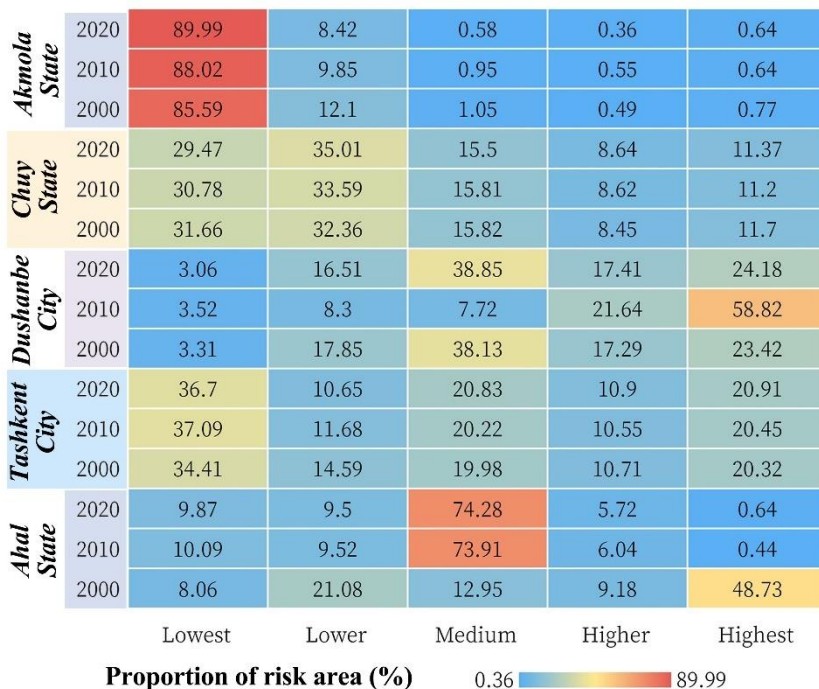

**Figure 8.** Proportion of different grades of ecological risk areas.

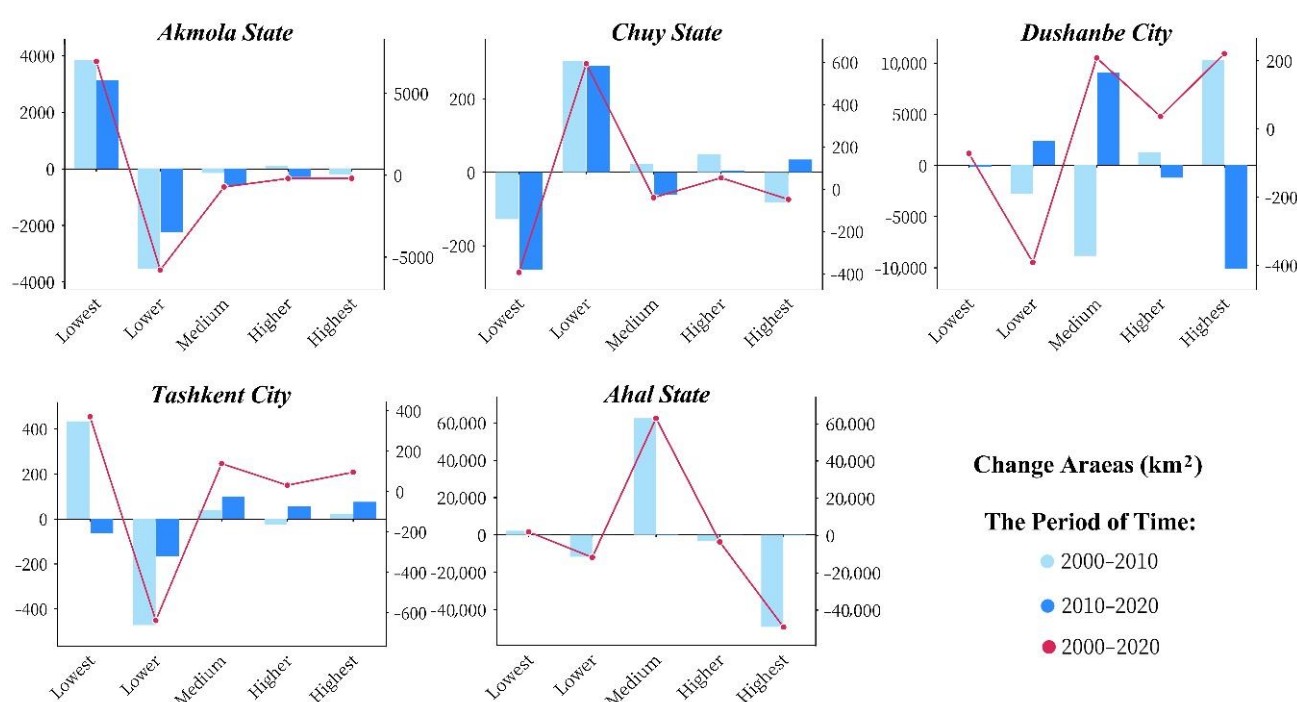

**Figure 9.** Changes of ecological risk area in different periods.

## 4. Discussion

### 4.1. Causes of Land Use Change

The cultivated land area of Akmola State decreased from 2000 to 2010 but increased from 2010 to 2020. As the largest grain producer in Central Asia and an important global grain exporter, Kazakhstan has a great potential for agricultural development. However, owing to the long-time low production level of agricultural machinery and equipment, the trade of agricultural products has developed at a lumbering pace [58]. Under the background of the BRI, China and Kazakhstan have continuously deepened cooperation in

agriculture, including the establishment of agricultural demonstration zones and agricultural free-trade zones. Since 2014, the Kazakh government has continuously launched a series of effective measures to encourage agricultural development, such as encouraging the renewal of agricultural machinery and equipment [59], which not only provides great opportunities for China's agricultural trade cooperation, but also greatly promotes the development of Kazakhstan's agricultural industry.

The water area of Chuy State increased significantly throughout the study period, particularly from 2000 to 2010. This is mainly because Kyrgyzstan is rich in alpine glacier resources, and many large cross-border rivers originate in China, allowing it to be fruitful in water resources. Glaciers and alpine snow have been melting due to climate change, resulting in melt runoff, which leads to an increase in the surface runoff area to a certain extent. However, glacier degradation inevitably leads to a series of serious ecological consequences such as debris flow and flood disasters [60,61]. Therefore, during the construction of the BRI we must adhere to the concept of ecological environmental protection and green and low-carbon development.

The water area of Dushanbe decreased significantly from 2000 to 2010, and the cultivated land area of Dushanbe decreased from 2010 to 2020. As the first country to sign a cooperation agreement with China on jointly building the Silk Road Economic Belt, Tajikistan has always actively responded to and supported the construction of the Silk Road Economic Belt. There are many mountains in the country, soil fertility is poor, and arable land accounts for only 6.1% of the land area. In addition, the reduction of agricultural investment leads to the abandonment of some agricultural land that is already in short supply, and the arable land area shows a decreasing trend [62–64]. Aligned with the BRI, although China and Tajikistan have made some progress in agricultural cooperation, the agricultural development of Tajikistan is still facing great challenges due to its poor agricultural resources and backward agricultural technology.

The area of cultivated and construction land in Tashkent City increased throughout the study period, and the area of grassland decreased. Uzbekistan has actively responded to the BRI. Increasing investment in agricultural infrastructure, improving agricultural processing technology, and expanding the scale of agricultural production. At the same time, the cooperation between China and Uzbekistan is mainly focused on the development of energy, mineral resources, and light industry. Those had promoted urbanization development and infrastructure construction to a great extent, and the characteristics of land use change are significant [65,66].

The grassland area of Ahal State decreased, and the unused land area increased from 2000 to 2010. The land use in Ahal State is mainly unused land and grassland, and the types of unused land are mainly sandy land and saline-alkali land. The climate in this region is dry, the temperature is high, and water evaporation loss is fast [67]. Concurrently, agricultural industrialization has caused great damage to soil structure, and the extensive use of mineral fertilizers has salinized soil [68]. Overgrazing destroys grassland, weakens wind and sand fixation, and strengthens the trend of land desertification. From 2010 to 2020, the grassland area of Ahal State decreased significantly, and the water area increased significantly. Ecological environmental protection is an important link in the sustainable development of the BRI. Countries within the Belt and Road have always adhered to the green "Belt and Road" concept in the process of cooperation and joint construction. In April 2017, the issuance of the guidance on promoting the construction of the green "Belt and Road" systematically elaborated the significance of building the green "Belt and Road" and fully integrated environmental protection and green concepts into the construction cooperation within the BRI [69]. In April 2019, the initiative of jointly building the development, sharing and outlook issued by the Chinese government summarized the phased achievements of the construction of the Belt and Road. The evolution and development of ecological and environmental protection policies have demonstrated China's determination to develop the green "Belt and Road" and systematically explained the importance and overall thinking of the construction of the green "Belt and Road."

*4.2. Dynamic Analysis of Ecological Risk*

Overall, the ecological risk level in the areas within the BRI is not high, but the proportion and change in ecological risk areas at different levels in different regions are quite different (Figures 8 and 9). See Appendix B (Tables A2–A6) for the landscape pattern indices of the five capitals.

Throughout the study period, the lowest ecological risk areas in Akmola State increased, the area of other ecological risk areas decreased, and the overall ecological risk level was low. During this period, the transfer mode of land use type in Akmola State was mainly from cultivated land to grassland. Cultivated land is a means of production formed by the transformation of natural soil through human agricultural production activities. Affected by scattered residential areas, patches were relatively broken, and the degree of human interference was large. Compared with cultivated land, grasslands have a better ecological environment, lower degree of landscape fragmentation, and lower ecological risk.

The areas of lower and higher ecological risk in Chuy State increased significantly from 2000 to 2010, mainly distributed at the edge of unused land in the middle, and the landscape fragmentation index and landscape separation index of forest land, grassland, and unused land were high. Simultaneously, the water area increased significantly during this period, and fragile landscape types led to an increase in ecological risk levels. From 2010 to 2020, the area of lowest ecological risk areas decreased significantly, and the area of lower ecological risk areas increased slightly. This is mainly due to the increase of landscape vulnerability and ecological risk level from forestland to grassland.

The area of medium ecological risk areas in Dushanbe City decreased from 2000 to 2010, mainly due to the significant reduction of water area, the reduction of landscape fragmentation, and the simultaneous reduction of landscape vulnerability. At the same time, the area of highest risk areas increased, mainly distributed in the staggered distribution area of unused land, grassland, and forest land in the West. The land type changed from forestland to grassland, and the degree of landscape vulnerability increased. From 2010 to 2020, the area of cultivated land and the degree of human disturbance decreased. Concurrently, the grassland landscape separation index and landscape fragmentation index in the region were small, the area of the highest ecological risk area decreased, and the area of the lower ecological risk area increased.

The area of the lowest ecological risk area in Tashkent City decreased and the area of the highest ecological risk area increased throughout the study period. The overall change from 2010 to 2020 was greater than that in the previous period. The conversion of grassland to forestland in Tashkent shows that the ecological environment is developing better, the ability of landscape types to resist external interference is improved, and the level of ecological risk is reduced. The conversion of grassland to unused land has deteriorated the quality of the regional ecological environment. At the same time, the unused land was obviously affected by the natural environment, the landscape vulnerability index was the largest, and the area of the highest ecological risk area increased.

The area of the highest risk areas in Ahal State decreased significantly from 2000 to 2010. During this period, the degree of landscape fragmentation of forestland and unused land decreased, and the separation index and landscape interference index of unused land decreased significantly, indicating that the patch distribution of unused land was relatively concentrated, and the degree of landscape interference was relatively low; thus, the area of the highest risk area was reduced. From 2010 to 2020, the change in ecological risk areas at all levels was basically stable, and the ecological risk areas were mainly medium ecological risk areas.

The overall ecological risk of Akmola State was low, and the area of the lowest risk area increased during the study period. The main reason was that farmland was converted to grassland, the ecological environment was better, and the degree of landscape fragmentation was lower. Therefore, the objective of ecological risk regulation in Akmola State is to optimize the land use structure, give full play to the ecological service function of

land and provide comprehensive benefits of land use without changing the regional land-use pattern. In Chuy State and Dushanbe City, the highest ecological risk increased, mainly distributed in the interlacing zone of unused land, forest land and grassland. Therefore, the objective of ecological risk regulation is to adjust the land use structure and supplement more ecological land in the staggered region. Tashkent City and Ahal State should optimize the land use structure and establish a stable ecological security pattern of land use without changing the regional land-use pattern.

## 5. Conclusions

Based on land use data, this study systematically analyzed the dynamic changes in land use and land type transfer patterns of key regions in "Belt and Road" countries. The evaluation model of landscape ecological risk index was constructed, the ecological risk level of five cities and states was evaluated, and the temporal and spatial variation of ecological risk was analyzed. The main conclusions of this study are as follows:

First, the landscape types of the five cities and states were mainly grassland, cultivated land, and unused land. The landscape types that changed significantly during the study period were water area and unused land. In Chuy State, the dynamic degree of water increased, with the largest being 8.29%. Dushanbe City had a maximum dynamic reduction of water and unused land of 2.53% and 0.09%, respectively. The maximum dynamic growth rate of unused land in Akmola State was 10.88%. The construction land areas of the five capitals show a trend of continuous increase. Second, the transfer direction of land-use types in the five capitals is mainly between cultivated land and grassland, grassland and forestland, grassland, and unused land. Akmola State is mainly converted from cultivated land to grassland, Chuy State is mainly converted from forestland to grassland, Dushanbe City and Tashkent City are mainly converted from grassland to forestland, and Ahal State is mainly converted from grassland to unused land. Third, overall, the distribution of the highest risk areas and the higher risk areas is consistent with that of unused land and water. The medium risk areas were mainly distributed in the unused land, the river at the edge of the cultivated land, and other landscape distribution areas. The distribution of the lowest-risk area and lower risk area is consistent with that of cultivated land, forestland, grassland, and construction land. The landscape fragmentation and vulnerability indices have a significant influence on the ecological risk level. Furthermore, from the area proportion and change of different levels of ecological risk areas, Akmola State had the largest proportion of lowest ecological risk areas and the smallest proportion of highest ecological risk areas. The areas with the highest ecological risk in Chuy State and Dushanbe City showed an increasing trend. The higher ecological risk areas and highest ecological risk areas in Tashkent City increased slightly, and the ecological risk level was stable during the study period. In 2020, from the area proportion and change of highest and higher ecological risk areas, Dushanbe City has the highest proportion of 41.59%, Tashkent City 31.81%, and Akmola State only 1%.

This paper provides a practical study on land use change in key areas within the BRI and provides a grid-scale landscape ecological risk assessment. With the further promotion and implementation of the BRI, rapid urbanization, infrastructure projects, more complex land-use patterns and structures, and increasingly severe impacts on the ecological environment, more research practices are urgently needed to provide a research basis for the development of countries and regions within the BRI. However, in this study, only the changes of land use and ecological risks in the 10 years before and after the BRI were compared. With the growth of the implementation cycle of the BRI, the comparison of long-term impacts before and after is unclear. With the further implementation of the BRI, future research should also focus on the changes in land use and ecological risks caused by long-term policy implementation. On the other hand, the study area selected in this study is representative for reflecting the land use change and ecological risk in Central Asia under the background of the BRI, but it is not universal for other regions, and relevant studies need to be further strengthened in future studies.

**Author Contributions:** Conceptualization, X.Z.; methodology, X.Z. and L.Y.; software, L.Y.; validation, X.Z.; formal analysis, X.Z. and L.Y.; investigation, X.Z., L.Y. and J.L.; resources, X.Z. and J.L.; data curation, X.Z., L.Y. and J.L. writing—original draft preparation, X.Z., L.Y. and W.L.; writing—review and editing, X.Z., L.Y., J.L. and W.L.; visualization, X.Z. and J.L.; supervision, X.Z. and W.L.; project administration, X.Z.; funding, X.Z. All authors have read and agreed to the published version of the manuscript.

**Funding:** This research was funded by the National Natural Science Foundation of China (No. 42101276), Science and technology project of Gansu Province (No. 20JR5RA529) and National Natural Science Foundation of China (No. 41661035).

**Data Availability Statement:** The data presented in this study are available on request from the author. Most of the data can be obtained by visiting the following address: http://www.globallandcover.com/, accessed on 3 December 2021.

**Conflicts of Interest:** The authors declare no conflict of interest.

## Appendix A

**Table A1.** Land use type reclassification.

| Old Code | Old Type | Content | New Code | New Type |
|---|---|---|---|---|
| 10 | Cultivated land | Land used for planting crops, including paddy fields, irrigated dry land, rainfed dry land, vegetable fields, grass planting land, greenhouse land, land with fruit trees and other economic trees between planting crops, as well as tea gardens, coffee gardens and other shrub economic crops planting land. | 01 | Cultivated land |
| 20 | Forestland | Land covered with trees with canopy coverage of more than 30% includes deciduous broad-leaved forests, evergreen broad-leaved forests, deciduous coniferous forests, evergreen coniferous forests, mixed forests, and open woodlands with canopy coverage of 10–30%. | 02 | Forestland |
| 40 | Shrubland | Land covered by shrubby with shrub coverage greater than 30%, including montane, deciduous and evergreen, and desert areas with shrub coverage greater than 10%. | | |
| 30 | Grassland | Land covered by natural herbaceous vegetation with coverage greater than 10%, including steppe, meadow, savanna, desert steppe, and urban artificial grassland. | 03 | Grassland |
| 50 | Wetland | Located in the boundary zone between land and water, there is shallow water or soil too wet land, more growth of marsh or wet plants. Including inland marshes, lake marshes, river flooding wetlands, forest/bush wetlands, peat bogs, mangroves, salt marshes, etc. | 04 | Water |
| 60 | Water body | Land area covered by liquid water, including rivers, lakes, reservoirs, pits, etc. | | |
| 100 | Glaciers and permanent snow cover | Land covered by permanent snow, glaciers, and ice caps, including mountain snow, glaciers, and polar ice caps. | | |
| 80 | Artificial surface | The surface formed by artificial construction activities includes all kinds of residential land, industrial and mining facilities, transportation facilities, etc., excluding the contiguous green land and water bodies inside the construction land. | 05 | Construction land |
| 70 | Tundra | Land covered by lichens, mosses, hardy perennial herbs and shrubs in cold zone and alpine environment, including shrub tundra, grassland tundra, wet tundra, alpine tundra, bare tundra, etc. | 06 | Unused land |
| 90 | Bare land | Natural covered land with vegetation coverage less than 10%, including desert, sand, gravel, bare rock, saline-alkali land, etc. | | |

## Appendix B

**Table A2.** Landscape pattern index of Akmola State from 2000 to 2020.

| Land Type | Year | Area/k$^2$ | Number | Fragmentation | Abruption | Predominance | Obstruction | Fragility | Damnify |
|---|---|---|---|---|---|---|---|---|---|
| Cultivated land | 2000 | 80,247.20 | 1814 | 0.0002 | 0.0105 | 0.2068 | 0.0446 | 0.6000 | 0.0268 |
| | 2010 | 78,517.00 | 1802 | 0.0002 | 0.0107 | 0.2025 | 0.0438 | 0.6000 | 0.0263 |
| | 2020 | 79,294.08 | 2026 | 0.0003 | 0.0113 | 0.2052 | 0.0446 | 0.6000 | 0.0267 |
| Forestland | 2000 | 10,448.37 | 230,762 | 0.2208 | 0.9131 | 0.4481 | 0.4740 | 0.2000 | 0.0948 |
| | 2010 | 10,103.48 | 225,183 | 0.2229 | 0.9328 | 0.4587 | 0.4830 | 0.2000 | 0.0966 |
| | 2020 | 9135.92 | 209,662 | 0.2295 | 0.9956 | 0.4582 | 0.5051 | 0.2000 | 0.1010 |
| Grassland | 2000 | 61,560.11 | 67,997 | 0.0110 | 0.0841 | 0.2803 | 0.0868 | 0.4000 | 0.0347 |
| | 2010 | 64,629.88 | 63,697 | 0.0099 | 0.0776 | 0.2864 | 0.0855 | 0.4000 | 0.0342 |
| | 2020 | 61,803.83 | 59,042 | 0.0096 | 0.0781 | 0.2792 | 0.0840 | 0.4000 | 0.0336 |
| Water | 2000 | 4229.44 | 17,190 | 0.0406 | 0.6157 | 0.0421 | 0.2135 | 0.8000 | 0.1708 |
| | 2010 | 3138.66 | 10,893 | 0.0347 | 0.6605 | 0.0289 | 0.2213 | 0.8000 | 0.1770 |
| | 2020 | 5588.09 | 10,039 | 0.0180 | 0.3561 | 0.0350 | 0.1228 | 0.8000 | 0.0983 |
| Construction land | 2000 | 1089.79 | 899 | 0.0082 | 0.5462 | 0.0044 | 0.1689 | 0.0000 | 0.0000 |
| | 2010 | 1195.52 | 1040 | 0.0087 | 0.5358 | 0.0050 | 0.1661 | 0.0000 | 0.0000 |
| | 2020 | 1523.97 | 1651 | 0.0108 | 0.5296 | 0.0073 | 0.1658 | 0.0000 | 0.0000 |
| Unused land | 2000 | 210.40 | 9769 | 0.4638 | 9.3197 | 0.0184 | 3.0315 | 1.0000 | 3.0315 |
| | 2010 | 200.77 | 9356 | 0.4671 | 9.5904 | 0.0185 | 3.1143 | 1.0000 | 3.1143 |
| | 2020 | 439.41 | 6736 | 0.1534 | 3.7127 | 0.0151 | 1.1935 | 1.0000 | 1.1935 |

**Table A3.** Landscape pattern index of Chuy State from 2000 to 2020.

| Land Type | Year | Area/k$^2$ | Number | Fragmentation | Abruption | Predominance | Obstruction | Fragility | Damnify |
|---|---|---|---|---|---|---|---|---|---|
| Cultivated land | 2000 | 642,467.70 | 185 | 0.0003 | 0.0160 | 0.1128 | 0.0275 | 0.6000 | 0.0165 |
| | 2010 | 643,289.22 | 210 | 0.0003 | 0.0170 | 0.1130 | 0.0279 | 0.6000 | 0.0167 |
| | 2020 | 617,085.36 | 414 | 0.0007 | 0.0249 | 0.1090 | 0.0296 | 0.6000 | 0.0178 |
| Forestland | 2000 | 184,874.40 | 113,205 | 0.6123 | 1.3761 | 0.3192 | 0.7828 | 0.2000 | 0.1566 |
| | 2010 | 165,943.89 | 113,040 | 0.6812 | 1.5320 | 0.3177 | 0.8637 | 0.2000 | 0.1727 |
| | 2020 | 165,885.12 | 112,708 | 0.6794 | 1.5303 | 0.3154 | 0.8619 | 0.2000 | 0.1724 |
| Grassland | 2000 | 1,001,815.92 | 54,659 | 0.0546 | 0.1765 | 0.3137 | 0.1430 | 0.4000 | 0.0572 |
| | 2010 | 1,019,185.92 | 53,056 | 0.0521 | 0.1709 | 0.3137 | 0.1400 | 0.4000 | 0.0560 |
| | 2020 | 1,036,634.31 | 53,063 | 0.0512 | 0.1680 | 0.3161 | 0.1392 | 0.4000 | 0.0557 |
| Water | 2000 | 5446.26 | 3091 | 0.5675 | 7.7189 | 0.0088 | 2.6012 | 0.8000 | 2.0810 |
| | 2010 | 8172.00 | 2201 | 0.2693 | 4.3410 | 0.0070 | 1.4384 | 0.8000 | 1.1507 |
| | 2020 | 9961.83 | 2544 | 0.2554 | 3.8285 | 0.0082 | 1.2779 | 0.8000 | 1.0223 |
| Construction land | 2000 | 82,155.33 | 314 | 0.0038 | 0.1631 | 0.0152 | 0.0539 | 0.0000 | 0.0000 |
| | 2010 | 83,107.35 | 324 | 0.0039 | 0.1638 | 0.0154 | 0.0542 | 0.0000 | 0.0000 |
| | 2020 | 90,392.40 | 431 | 0.0048 | 0.1737 | 0.0169 | 0.0579 | 0.0000 | 0.0000 |
| Unused land | 2000 | 370,279.44 | 65,359 | 0.1765 | 0.5221 | 0.2304 | 0.2909 | 1.0000 | 0.2909 |
| | 2010 | 367,340.67 | 66,128 | 0.1800 | 0.5293 | 0.2331 | 0.2954 | 1.0000 | 0.2954 |
| | 2020 | 367,080.03 | 66,993 | 0.1825 | 0.5332 | 0.2344 | 0.2981 | 1.0000 | 0.2981 |

**Table A4.** Landscape pattern index of Dushanbe City from 2000 to 2020.

| Land Type | Year | Area/k$^2$ | Number | Fragmentation | Abruption | Predominance | Obstruction | Fragility | Damnify |
|---|---|---|---|---|---|---|---|---|---|
| Cultivated land | 2000 | 168,396.57 | 538 | 0.0032 | 0.1232 | 0.0215 | 0.0429 | 0.6000 | 0.0257 |
| | 2010 | 173,236.50 | 841 | 0.0049 | 0.1498 | 0.0224 | 0.0518 | 0.6000 | 0.0311 |
| | 2020 | 171,467.28 | 1194 | 0.0070 | 0.1803 | 0.0224 | 0.0620 | 0.6000 | 0.0372 |
| Forestland | 2000 | 428,648.04 | 327,580 | 0.7642 | 1.1944 | 0.3327 | 0.8070 | 0.2000 | 0.1614 |
| | 2010 | 426,887.82 | 324,858 | 0.7610 | 1.1944 | 0.3287 | 0.8046 | 0.2000 | 0.1609 |
| | 2020 | 426,930.12 | 327,358 | 0.7668 | 1.1989 | 0.3254 | 0.8081 | 0.2000 | 0.1616 |

**Table A4.** *Cont.*

| Land Type | Year | Area/k² | Number | Fragmentation | Abruption | Predominance | Obstruction | Fragility | Damnify |
|---|---|---|---|---|---|---|---|---|---|
| Grassland | 2000 | 1,565,196.75 | 166,378 | 0.1063 | 0.2331 | 0.3374 | 0.1906 | 0.4000 | 0.0762 |
| | 2010 | 1,567,508.31 | 171,794 | 0.1096 | 0.2365 | 0.3415 | 0.1941 | 0.4000 | 0.0776 |
| | 2020 | 1,566,505.80 | 177,251 | 0.1132 | 0.2404 | 0.3430 | 0.1973 | 0.4000 | 0.0789 |
| Water | 2000 | 26,459.01 | 3480 | 0.1315 | 1.9944 | 0.0063 | 0.6653 | 0.8000 | 0.5323 |
| | 2010 | 18,471.24 | 2192 | 0.1187 | 2.2675 | 0.0042 | 0.7404 | 0.8000 | 0.5923 |
| | 2020 | 19,776.15 | 2109 | 0.1066 | 2.0774 | 0.0042 | 0.6774 | 0.8000 | 0.5419 |
| Construction land | 2000 | 44,562.60 | 333 | 0.0075 | 0.3663 | 0.0059 | 0.1148 | 0.0000 | 0.0000 |
| | 2010 | 46,161.09 | 339 | 0.0073 | 0.3568 | 0.0061 | 0.1119 | 0.0000 | 0.0000 |
| | 2020 | 47,721.51 | 442 | 0.0093 | 0.3941 | 0.0063 | 0.1241 | 0.0000 | 0.0000 |
| Unused land | 2000 | 967,535.82 | 205,773 | 0.2127 | 0.4194 | 0.2963 | 0.2914 | 1.0000 | 0.2914 |
| | 2010 | 968,802.66 | 207,851 | 0.2145 | 0.4210 | 0.2972 | 0.2930 | 1.0000 | 0.2930 |
| | 2020 | 968,730.30 | 213,646 | 0.2205 | 0.4268 | 0.2986 | 0.2980 | 1.0000 | 0.2980 |

**Table A5.** Landscape pattern index of Tashkent City from 2000 to 2020.

| Land Type | Year | Area/k² | Number | Fragmentation | Abruption | Predominance | Obstruction | Fragility | Damnify |
|---|---|---|---|---|---|---|---|---|---|
| Cultivated land | 2000 | 545,191.74 | 246 | 0.0005 | 0.0188 | 0.1277 | 0.0314 | 0.6000 | 0.0189 |
| | 2010 | 564,322.05 | 334 | 0.0006 | 0.0212 | 0.1325 | 0.0331 | 0.6000 | 0.0199 |
| | 2020 | 565,452.81 | 576 | 0.0010 | 0.0278 | 0.1332 | 0.0355 | 0.6000 | 0.0213 |
| Forestland | 2000 | 316,866.78 | 113,666 | 0.3587 | 0.6965 | 0.3102 | 0.4504 | 0.2000 | 0.0901 |
| | 2010 | 312,843.42 | 108,056 | 0.3454 | 0.6877 | 0.3082 | 0.4407 | 0.2000 | 0.0881 |
| | 2020 | 313,368.93 | 110,987 | 0.3542 | 0.6959 | 0.3076 | 0.4474 | 0.2000 | 0.0895 |
| Grassland | 2000 | 462,884.49 | 64,896 | 0.1402 | 0.3603 | 0.2429 | 0.2268 | 0.4000 | 0.0907 |
| | 2010 | 445,574.43 | 63,224 | 0.1419 | 0.3694 | 0.2416 | 0.2301 | 0.4000 | 0.0920 |
| | 2020 | 444,774.51 | 66,019 | 0.1484 | 0.3782 | 0.2433 | 0.2363 | 0.4000 | 0.0945 |
| Water | 2000 | 12,784.32 | 3182 | 0.2489 | 2.8884 | 0.0096 | 0.9929 | 0.8000 | 0.7943 |
| | 2010 | 13,059.27 | 2745 | 0.2102 | 2.6259 | 0.0090 | 0.8947 | 0.8000 | 0.7157 |
| | 2020 | 14,803.92 | 3109 | 0.2100 | 2.4656 | 0.0100 | 0.8467 | 0.8000 | 0.6773 |
| Construction land | 2000 | 156,381.21 | 568 | 0.0036 | 0.0998 | 0.0377 | 0.0393 | 0.0000 | 0.0000 |
| | 2010 | 160,309.98 | 572 | 0.0036 | 0.0976 | 0.0387 | 0.0388 | 0.0000 | 0.0000 |
| | 2020 | 157,126.95 | 765 | 0.0049 | 0.1152 | 0.0383 | 0.0447 | 0.0000 | 0.0000 |
| Unused land | 2000 | 219,983.13 | 106,096 | 0.4823 | 0.9693 | 0.2719 | 0.5863 | 1.0000 | 0.5863 |
| | 2010 | 217,510.02 | 100,762 | 0.4633 | 0.9552 | 0.2701 | 0.5722 | 1.0000 | 0.5722 |
| | 2020 | 218,564.55 | 102,510 | 0.4690 | 0.9589 | 0.2676 | 0.5757 | 1.0000 | 0.5757 |

**Table A6.** Landscape pattern index of Ahal State from 2000 to 2020.

| Land Type | Year | Area/k² | Number | Fragmentation | Abruption | Predominance | Obstruction | Fragility | Damnify |
|---|---|---|---|---|---|---|---|---|---|
| Cultivated land | 2000 | 168,396.57 | 538 | 0.0032 | 0.1232 | 0.0215 | 0.0429 | 0.6000 | 0.0257 |
| | 2010 | 173,236.50 | 841 | 0.0049 | 0.1498 | 0.0224 | 0.0518 | 0.6000 | 0.0311 |
| | 2020 | 171,467.28 | 1194 | 0.0070 | 0.1803 | 0.0224 | 0.0620 | 0.6000 | 0.0372 |
| Forestland | 2000 | 428,648.04 | 327,580 | 0.7642 | 1.1944 | 0.3327 | 0.8070 | 0.2000 | 0.1614 |
| | 2010 | 426,887.82 | 324,858 | 0.7610 | 1.1944 | 0.3287 | 0.8046 | 0.2000 | 0.1609 |
| | 2020 | 426,930.12 | 327,358 | 0.7668 | 1.1989 | 0.3254 | 0.8081 | 0.2000 | 0.1616 |
| Grassland | 2000 | 1,565,196.75 | 166,378 | 0.1063 | 0.2331 | 0.3374 | 0.1906 | 0.4000 | 0.0762 |
| | 2010 | 1,567,508.31 | 171,794 | 0.1096 | 0.2365 | 0.3415 | 0.1941 | 0.4000 | 0.0776 |
| | 2020 | 1,566,505.80 | 177,251 | 0.1132 | 0.2404 | 0.3430 | 0.1973 | 0.4000 | 0.0789 |
| Water | 2000 | 26,459.01 | 3480 | 0.1315 | 1.9944 | 0.0063 | 0.6653 | 0.8000 | 0.5323 |
| | 2010 | 18,471.24 | 2192 | 0.1187 | 2.2675 | 0.0042 | 0.7404 | 0.8000 | 0.5923 |
| | 2020 | 19,776.15 | 2109 | 0.1066 | 2.0774 | 0.0042 | 0.6774 | 0.8000 | 0.5419 |
| Construction land | 2000 | 44,562.60 | 333 | 0.0075 | 0.3663 | 0.0059 | 0.1148 | 0.0000 | 0.0000 |
| | 2010 | 46,161.09 | 339 | 0.0073 | 0.3568 | 0.0061 | 0.1119 | 0.0000 | 0.0000 |
| | 2020 | 47,721.51 | 442 | 0.0093 | 0.3941 | 0.0063 | 0.1241 | 0.0000 | 0.0000 |
| Unused land | 2000 | 967,535.82 | 205,773 | 0.2127 | 0.4194 | 0.2963 | 0.2914 | 1.0000 | 0.2914 |
| | 2010 | 968,802.66 | 207,851 | 0.2145 | 0.4210 | 0.2972 | 0.2930 | 1.0000 | 0.2930 |
| | 2020 | 968,730.30 | 213,646 | 0.2205 | 0.4268 | 0.2986 | 0.2980 | 1.0000 | 0.2980 |

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
