# Peer review of "Exploring Changes in Land Use and Landscape Ecological Risk in Key Regions of the Belt and Road Initiative Countries"

_land, doi:10.3390/land11060940_

Round 1

Reviewer 1 Report

There are some serious problems with this paper. First, the content of the Belt and Road policy in each city is unclear, and how it affected land use is unclear. The second is how to set the category of ecological risk. The authors have only statistically classified them and have not considered their ecological implications at all. Furthermore, no novelty is found in the analysis method.

Reviewer 2 Report

This manuscript deals with a relevant issue: the land use transformation  and the landscape ecological risk provoked by big infrastructure projects.

The authors constructed and applied an evaluation model of landscape ecological risk index to characterize and analyze the dynamic changes in land use and  landscape ecological risk in some countries in Central Asia that are members of the Belt and Road Initiative (BRI).

The research issue and objetives are clear and coherently stated.

The method is sound and coherent with the research objectives.

The results and discussion are clear and soundly stated.

The conclusions are supported by the research results.

Nevertheless, some points are worth revisiting:

1 - Into what extent the land use transformantion was provoked by the Belt and Road Initiative projects? Are there correlations between dinamic of land use transformation and the amount(US$) of investiment of the Belt and Road Initiative in the locals investigatred? What is the amount of BRI´s investiment in each local/city/country?

2 - Into what extent is coherent/consistent/reliable to compare Cities with States? Into what extent being a city or a state influenced the Level of Ecological Risk presented in Figure3, Figure 4  and Figure 5?

3 - In the section of Conclusions, it seems relevant and opportune to be a little more especific about the policy and managerial implications that can be envisaged from the research results. What are the research limitations?

4 - In line 195 it stated the word "obvious" and in line 198 it is stated the word "trivial". What "obvious" and "trivial" mean?

Reviewer 3 Report

What I see here is a relatively good and rather clear article that should be improved substantially before publication and for this reason cannot be published in the present form. I would see moderate/major revisions clarifying (i) the representativeness of the case study in a broader, possibly global context, and (ii) the operational framework estimating capitals, together with a broader and literature-oriented definition of capital (economic, social, environmental). I would see a broder literature review, too. I am not sure authors have discussed enough the originality and novelty of their approach, that seems to be quite routinary and well established in literature, but not so much novel and original to merit a publication in a top-level journal. However, authors have room to clarify this point and I have nothing against future publication of this article. Thank you if you embark in a revision following my advice.

Round 2

Reviewer 1 Report

I think the author responds appropriately to the issues pointed out.

Reviewer 3 Report

Rather good revisions overall. Compliments to authors.

This manuscript is a resubmission of an earlier submission. The following is a list of the peer review reports and author responses from that submission.